# Attention-Only Transformers via Unrolled Subspace Denoising

## Abstract

Despite the great success of transformers in practice, their architectures have been empirically designed, hence lack of mathematical justification and interpretability. Moreover, many empirical studies have indicated that some components of the transformer architectures may be redundant and can be removed or replaced without compromising overall performance. Hence to derive a compact and interpretable transformer architecture, we contend that the goal of representation learning is to compress a set of noisy initial token representations towards a mixture of low-dimensional subspaces. Based on the existing literature, the associated denoising operation naturally takes the form of a multi-subspace self-attention (MSSA). By unrolling such iterative denoising operations as a deep network, we arrive at a highly compact architecture that consists of only an MSSA operator with skip connections at each layer, without MLP. We rigorously prove that each layer of the proposed transformer performs so highly efficient denoising that it improves the signal-to-noise ratio of token representations *at a linear rate* with respect to the number of layers. Despite its simplicity, extensive experiments on language and vision tasks demonstrate that such a minimalistic attention-only transformer can achieve performance close to conventional transformers, such as GPT-2 and CRATE.

## 1 Introduction

Over the past years, transformer architectures (Vaswani et al., 2017) have achieved remarkable empirical success across various modern machine learning applications, including large language models (LLMs) (Devlin, 2018; Brown et al., 2020a), vision generative models (Chen et al., 2020; Bao et al., 2023; Peebles & Xie, 2023), and reinforcement learning (Chen et al., 2021). In general, transformer architectures are constructed by stacking multiple identical layers that work together to process and learn from data. Each layer is composed of several interacting components arranged in a specific sequence, including self-attention operators, layer normalization, multilayer perceptron (MLP) networks, and skip connections. In practice, transformer architectures, such as BERT (Devlin, 2018) and GPT-4 (Achiam et al., 2023), are highly deep, often with dozens to even hundreds of layers, and are significantly over-parameterized, containing millions or even billions of parameters. This considerable depth and a large number of parameters endow transformers with impressive learning capabilities, allowing them to model complex patterns and relationships in real-world data.

Despite the remarkable success of transformers, their deep and over-parameterized architecture makes them complex "black box", hindering interpretability and the understanding of their inner mechanism. To address this, a common approach involves systematically removing or modifying certain components in transformers to simplify the architecture; see, e.g., Dong et al. (2021); Alcalde et al. (2024); Noci et al. (2024); Geshkovski et al. (2023a); Geva et al. (2020); Guo et al. (2024). For example, Alcalde et al. (2024) studied pure-attention hard-max transformers with skip connections and showed that the output converges to a clustered equilibrium as the number of layers goes to infinity. Noci et al. (2024) analyzed a modified softmax-based attention model with skip connections, demonstrating that the limiting distribution can be described by a stochastic differential equation. These studies indicate that the most basic components of transformers are self-attention layers and skip connections. Although existing studies have provided valuable insights into different components of transformers, few of them elucidate the underlying mechanisms by which transformers process and transform input into output across layers.

Moreover, existing empirical studies suggest that some components of transformers are not be essential and can be removed or modified without compromising performance. For example, He & Hofmann (2024) empirically demonstrated that transformer architecture can be simplified by removing components such as skip connections, value matrix, and normalization layers without degrading performance. Additionally, Sukhbaatar et al. (2019) investigated the effects of removing MLP blocks from transformers and augmenting the self-attention layers to play a similar role to MLP blocks, showing that performance can be preserved. Similarly, Pires et al. (2023) examined the potential for reducing the frequency of MLP layers in transformers. Other works also studied other simplifications of transformers, such as linear attentions (Katharopoulos et al., 2020) and shared-QK attentions (Kitaev et al., 2020). Based on these discussions, this work focuses on addressing the following question regarding the understanding of the underlying mechanism of transformers and the design of their architectures:

*Can we design a minimalistic transformer-like deep architecture consisting of fully interpretable and provably effective layers that achieves performance close to that of standard transformers?*

## 1.1 RELATED WORKS

**Existing studies on self-attention mechanisms.** It is widely believed that the power of transformers primarily stems from their self-attention layers, which enable the model to capture long-range dependencies and contextual relationships between tokens by dynamically weighing token relationships across the input sequence (Tsai et al., 2019; Vaswani et al., 2017). To explore the mechanism behind self-attention, numerous studies have investigated the performance of pure self-attention networks, often incorporating only one additional component to prevent rank collapse and maintain expressiveness. For example, Dong et al. (2021) showed that in pure-attention transformers without skip connections and MLP layers, token representations collapse exponentially to a rank-1 matrix across layers. They also showed that self-attention networks with skip connections prevent rank collapse. Geshkovski et al. (2023a;b) studied the dynamics of multi-head self-attentions and characterized clustering behaviors of learned representations. Recently, Wu et al. (2024) showed that pure self-attention networks with LayerNorm can prevent rank collapse. While these studies have advanced the theoretical understanding of self-attention mechanisms in simplified transformer architectures, they don't provide any empirical validation on real-world vision or language tasks, offering little insight into the role of self-attention in practice.

**Deep network architecture design via unrolled optimization.** It is commonly believed that the success of modern deep networks largely stems from their ability to transform the raw data into compact and structured representations, which facilitates downstream tasks (Chan et al., 2022; Chen et al., 2023; Ma et al., 2022; Yu et al., 2023a). A principled and interpretable approach to learning such representations with transformers is to construct an architecture that incrementally transforms tokens into these representations via unrolling iterative optimization steps as layers of a deep network (Chan et al., 2022; Monga et al., 2021; Wang et al., 2016; Yu et al., 2023b; Zhang & Ghanem, 2018). Notably, Monga et al. (2021) demonstrate that such unrolled networks are more interpretable, parameter-efficient, and effective compared to generic networks. In this approach, each iteration of an algorithm for learning compact and structured representations is represented as one layer of deep networks. For example, Gregor & LeCun (2010) have demonstrated that sparse coding al-

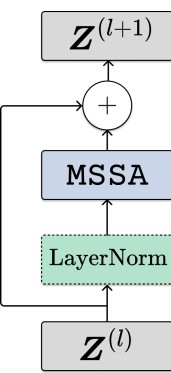

Figure 1: **Each layer of the proposed attention-only transformer architecture.**

gorithms, such as ISTA, can be used to construct MLPs. Recently, Chan et al. (2022) constructed a "white-box" network based on an iterative gradient descent scheme to optimize the maximal coding rate reduction objective. More recently, Yu et al. (2023a) designed a "white-box" transformer architecture by implementing an approximate alternating minimization to optimize the sparse rate reduction objective. The proposed transformer achieves performance comparable to some popular ones such as ViT (Dosovitskiy et al., 2020), BERT (Devlin, 2018), and DINO (Caron et al., 2021) on vision tasks. Notably, a key component in their design is the multi-head subspace self-attention (MSSA) operator (see Eq. (3)). While they argued that this operator can denoise token representations, they only showed that the negative gradient of the compression term of the objective points

to the denoising direction, without providing an accurate analysis or guarantee for the denoising efficiency. The MSSA's denoising capabilities remain an open question.

**Linear representation & superposition hypotheses.** Recent empirical studies on language tasks have raised the "linear representation hypothesis", which posits that token representations can be linearly encoded as one-dimensional feature vectors in the activation space of LLMs (Jiang et al., 2024; Park et al., 2023), and "superposition hypothesis", which further hypothesizes that token representations are a sparse linear combination of these feature vectors (Elhage et al., 2022; Yun et al., 2021; Arora et al., 2018). Building on these hypotheses, various approaches have been proposed to understand and utilize token representations. For example, Templeton (2024) employed sparse autoencoders to decompose the token representations of Claude 3 Sonnet into more interpretable pieces. Luo et al. (2024) leveraged sparse dictionary learning to explore token representations, decomposing them into interpretable components based on a concept dictionary. Recently, Engels et al. (2024) conjectured that token representations in LLMs are the sum of many sparse multi-dimensional features. This conjecture is supported by their experiments on GPT-2 and Mistral 7B, where they used sparse autoencoders to identify multi-dimensional features. Notably, all of these empirical studies come to the qualitative conclusion that *the token representations lie on a union of (possibly many) low-dimensional subspaces*.

### 1.2 OUR CONTRIBUTIONS

Based on the above discussions, we use a simple yet effective model for the token representations that accurately reflects the behaviors of trained transformers (such as LLMs) based on the previously referenced empirical studies. That is, we model the underlying distribution of token representations as a mixture of low-rank Gaussians corrupted by noise (see Definition 1). Specifically, each token representation lies in a subspace corrupted by the noise from other spaces (see Eq. (1)). To denoise these token representations, we employ the multi-head subspace self-attention (MSSA) operator proposed in (Yu et al., 2023a; Pai et al., 2023) to incrementally update the token representations (see Eq. (3)). Then, our contributions can be summarized as follows:

- **Attention-only transformer with a minimalistic architecture via unrolled optimization.** Based on unrolling the iterative optimization steps Eq. (3), we construct a new transformer with a streamlined architecture, consisting of only MSSA layers with skip connections (see Figure 1).[1] This design simplifies transformer architectures significantly compared to standard decoder-only transformers. More details are illustrated in Figure 3.

- **Theoretical guarantees on the denoising performance of the proposed transformer.** To quantify the denoising performance, we define a signal-to-noise (SNR) metric (see Eq. (8)) for each block of the token representations. We prove that each layer of the proposed transformer improves the SNR at a linear rate when the initial token representations are sampled from a mixture of low-rank Gaussians (see Theorem 1). This indicates the MSSA operator is highly effective in denoising token representations towards their corresponding subspaces.

- **Understanding roles of self-attention and MLP layers.** Notably, the proposed transformer is a valuable model for understanding the mechanism of attention since it disentangles the effect of MLP layers. Moreover, comparing the proposed transformer to standard transformers provides insights into the specific role, or empirical benefits, of the MLP layers in different tasks, such as for in-context learning (see experiments in Section 4.1.2).

We have conducted extensive experiments on both language and vision tasks, including causal language modeling, in-context learning, and supervised image classification, to complement our theory and demonstrate the potential of our proposed transformer architecture. These experiments highlight its ability to handle complex real-world applications, thereby confirming the practical value of our streamlined attention-only transformer design.

**Notation.** Given an integer $n$, we denote by $[n]$ the set $\{1, \ldots, n\}$. Given a vector $\boldsymbol{a}$, let $\|\boldsymbol{a}\|$ denote the Euclidean norm of $\boldsymbol{a}$ and $\mathrm{diag}(\boldsymbol{a})$ denote the diagonal matrix with $\boldsymbol{a}$ as its diagonal. Given a matrix $\boldsymbol{A}$, let $\|\boldsymbol{A}\|$ denote the spectral norm of $\boldsymbol{A}$, $\|\boldsymbol{A}\|_F$ denote the Frobenius norm, and $a_{ij}$ denote the $(i, j)$-th element. For sequences of positive numbers $\{a_n\}$ and $\{b_n\}$, we write $a_n \lesssim b_n$ or $b_n \gtrsim a_n$ if there exists an absolute constant $C > 0$ such that $a_n \leq Cb_n$. Given a constant $\tau > 0$, we define $\mathbb{I}(x > \tau) = 1$ if $x > \tau$ and $\mathbb{I}(x > \tau) = 0$ otherwise.

---

[1]For language tasks, we additionally include LayerNorm layers to improve performance.

## 2 TECHNICAL APPROACH AND JUSTIFICATION

To begin, we introduce the basic setup of transformers for learning representations from real-world data. Real-world data, such as images, videos, and text, are often modeled as random samples drawn from a high-dimensional probability distribution with low-dimensional intrinsic structures (Pope et al., 2020; Wright & Ma, 2022). Instead of directly inputting data samples into transformers, a common preprocessing step involves converting each sample into a sequence of vectors, referred to as tokens. Each token represents a localized segment of the data, such as an image patch, a snippet of text, or a frame in a video. Consequently, the input to transformers is typically a sequence of tokens, denoted as $\boldsymbol{X} = [\boldsymbol{x}_1, \ldots, \boldsymbol{x}_N] \in \mathbb{R}^{D \times N}$. Then, the goal of transformers is to learn a map $f : \mathbb{R}^{D \times N} \to \mathbb{R}^{d \times N}$ that transforms these tokens into structured and compact token representations that facilitate downstream tasks, such as classification (Dosovitskiy et al., 2020), segmentation (Kirillov et al., 2023), and generation (Saharia et al., 2022), by capturing the underlying patterns and relationships in the data. For ease of exposition, we denote the token representations as $\boldsymbol{Z} := f(\boldsymbol{X}) \in \mathbb{R}^{d \times N}$.

### 2.1 LEARNING TOKEN REPRESENTATIONS VIA UNROLLED OPTIMIZATION

In this subsection, we introduce how to learn token representations based on the approach of unrolling optimization algorithms (Chan et al., 2022; Gregor & LeCun, 2010; Monga et al., 2021; Sun et al., 2019; Wang et al., 2016; Yu et al., 2023b; Zhang & Ghanem, 2018). This approach involves constructing each layer of a neural network according to a step of an iterative optimization algorithm. That is, the network's architecture is designed to implement a specific optimization algorithm, where each layer corresponds to a single iterative step. By unrolling the algorithm, a "white-box" transformer architecture can be constructed as a multi-layer neural network that incrementally transforms input tokens into structured and compact representations. This process can be described as follows:

$$f : \boldsymbol{X} \xrightarrow{f^0} \boldsymbol{Z}^{(0)} \xrightarrow{f^1} \cdots\cdots \xrightarrow{f^l} \boldsymbol{Z}^{(l)} \xrightarrow{f^{l+1}} \cdots\cdots \xrightarrow{f^L} \boldsymbol{Z}^{(L)} = \boldsymbol{Z},$$

where $f^0 : \mathbb{R}^{D \times N} \to \mathbb{R}^{d \times N}$ is a pre-processing mapping (e.g., positional encoding, token embedding) that transforms input tokens $\boldsymbol{X} \in \mathbb{R}^{D \times N}$ to initial token representations $\boldsymbol{Z}^{(0)} \in \mathbb{R}^{d \times N}$, $f^l : \mathbb{R}^{d \times N} \to \mathbb{R}^{d \times N}$ denotes an incremental operation, and $\boldsymbol{Z}^{(l)}$ denotes the token representations at the $l$-th layer for each $l \in [L]$. Then, a key question is how to design the operator $f^l$ at each layer to learn meaningful token representations efficiently throughout the network in a principled manner.

### 2.2 DENOISING OPERATOR FOR LEARNING TOKEN REPRESENTATIONS

In this subsection, we introduce a denoising operator for learning token representations incrementally. To clarify the intuition behind this design, we assume that the initial token representations $\boldsymbol{Z}^{(0)}$ are drawn from a mixture of noisy low-rank Gaussian distributions as follows.

**Definition 1.** *Let $C_1, \ldots, C_K$ be a partition of the index set $[N]$ and $\boldsymbol{U}_k \in \mathbb{R}^{d \times p_k}$ denote the orthonormal basis of the $k$-th cluster for each $K \in [K]$. We say that the token representations $\{\boldsymbol{z}_i^{(0)}\}_{i=1}^N$ are sampled from a mixture of noisy low-rank Gaussian distributions if for each $k \in [K]$,*

$$\boldsymbol{z}_i^{(0)} = \boldsymbol{U}_k \boldsymbol{a}_i + \sum_{j \neq k}^K \boldsymbol{U}_j \boldsymbol{e}_{i,j}, \quad \forall i \in C_k, \tag{1}$$

*where $\boldsymbol{a}_i \overset{i.i.d.}{\sim} \mathcal{N}(\boldsymbol{0}, \boldsymbol{I}_{p_k})$ and $\boldsymbol{e}_{i,j} \overset{i.i.d.}{\sim} \mathcal{N}(\boldsymbol{0}, \delta^2 \boldsymbol{I}_{p_j})$ for all $i \in C_k$ and $k \in [K]$, $\{\boldsymbol{a}_i\}$ and $\{\boldsymbol{e}_{i,j}\}$ are respectively mutually independent, and $\{\boldsymbol{a}_i\}$ is independent of $\{\boldsymbol{e}_{i,j}\}$.*

Before proceeding, we make some remarks on this model. First, it provides a probabilistic framework for modeling token representations, assuming that they are sampled from a mixture of multiple low-rank Gaussian distributions with noise. Specifically, if a token representation belongs to the $k$-th cluster as shown in Eq. (1), it consists of a signal component $\boldsymbol{U}_k \boldsymbol{a}_i$ and a noise component $\sum_{j \neq k}^K \boldsymbol{U}_j \boldsymbol{e}_{i,j}$. Second, this model aligns well with the "linear representation hypothesis" (Jiang et al., 2024; Park et al., 2023) and "superposition hypothesis" (Elhage et al., 2022; Yun et al., 2021; Arora et al., 2018) regarding the structures of token representations in pretrained LLMs. Indeed, the

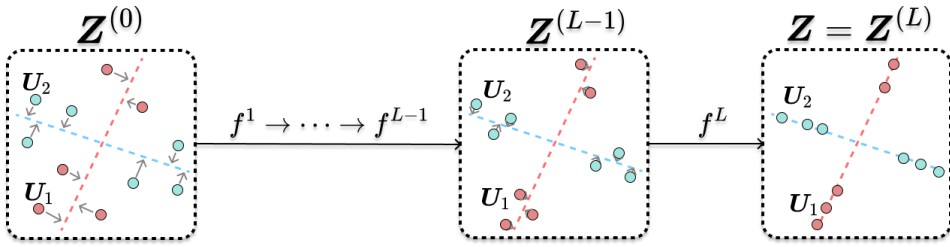

Figure 2: **Layers of transformers $f^l$ gradually denoise token representations towards their corresponding subspaces.**

bases of subspaces can be interpreted as semantics features, and each token representation can be approximately expressed as a sparse linear combination of subspace bases when the noise variance $\delta$ is sufficiently small. Our goal is to denoise these token representations towards the corresponding subspace; see Figure 2.

**Denoising operator for token representations.** In this work, we make the simplifying assumption that the subspaces are orthogonal to each other in Definition 1, i.e., $U_k^T U_j = 0$ for all $k \neq j$. Note this assumption is not so limiting as in high-dimensional spaces, with high-probability low-dimensional subspaces are incoherent, i.e., $U_k^T U_j \approx 0$ to each other (Wright & Ma, 2022).[2]

Without loss of generality, we rearrange the token representations $Z^{(0)}$ such that the token representations from the same subspace are concatenated together, i.e.,

$$Z^{(0)} = \begin{bmatrix} Z_1^{(0)} & \dots & Z_K^{(0)} \end{bmatrix} = \begin{bmatrix} U_1 A_1 + \sum_{j \neq 1} U_j E_{1,j} & \dots & U_K A_K + \sum_{j \neq K} U_j E_{K,j} \end{bmatrix},$$

where the columns of $Z_k^{(0)}$ denote the token representations from the $k$-th subspace for each $k \in [K]$, the columns of $A_k \in \mathbb{R}^{p_k \times N_k}$ consists of $\{a_i\}_{i \in C_k}$, and the columns of $E_{k,j} \in \mathbb{R}^{p_j \times N_k}$ consists of $\{e_{i,j}\}_{i \in C_k}$ for each $k \in [K]$ with $N_k = |C_k|$ for each $k \in [K]$. Obviously, projecting token representations onto their corresponding subspace helps separate the signal from the noise components, i.e.,

$$U_k U_k^T Z_s^{(0)} = \begin{cases} U_k A_k, & \text{if } s = k, \\ U_k E_{s,k}, & \text{if } s \neq k. \end{cases} \tag{2}$$

To denoise the token representations from $k$-th subspace, we can compute the similarity of projected token representations via $(U_k^T Z)^T (U_k^T Z)$ and verify that the similarity between projected token representations from the $k$-th subspace is high, while the similarity between other pairs of projected token representations is low when $\delta < 1$. Then, we convert it to a distribution of membership with function $\varphi$, such as hard-thresholding or soft-max functions, and denoise the token representations towards to the corresponding subspace using this membership. Now, we formalize the considered operator as follows:

$$Z^{(l+1)} = Z^{(l)} + \eta \sum_{k=1}^{K} U_k U_k^T Z^{(l)} \varphi\left(Z^{(l)^T} U_k U_k^T Z^{(l)}\right), \quad l = 0, 1, \dots, L-1, \tag{3}$$

where $\eta > 0$ is the denoising strength and $\varphi(\cdot) : \mathbb{R}^{d \times N} \to \mathbb{R}^{d \times N}$ is an operator applied to each column of an input matrix, i.e.,

$$\varphi(X) = [\varphi(x_1) \quad \dots \quad \varphi(x_N)]. \tag{4}$$

Notably, this operator, referred to as the *multi-head subspace self-attention* (**MSSA**), is first proposed by Yu et al. (2023a;b) to approximately optimize the compression term of the sparse rate reduction objective for constructing a transformer architecture. They showed that the negative compression gradient of the objective points from the token representation to the corresponding subspace. However, they do not give any accurate analysis of the denoising efficiency of the MSSA operator (3).

---

[2]It is not difficult to generalize our results to the more general case, with slightly more sophisticated analysis.

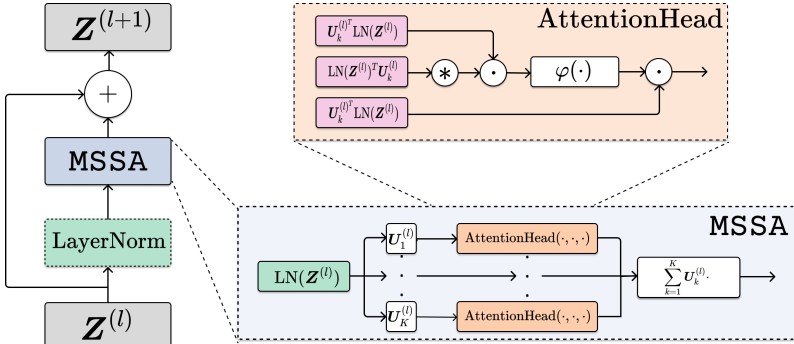

Figure 3: **Details of the attention-only transformer (AoT) architecture.** Each layer consists of the MSSA operator and a skip connection. In addition, LayeNnorm is included only for language tasks. In practice, backpropagation is applied to train the model parameters using training samples.

## 2.3 TRANSFORMER ARCHITECTURE DESIGN VIA UNROLLED OPTIMIZATION

Now, we formally introduce the proposed transformer architecture. Specifically, by unrolling the iterative optimization steps (3) as layers of a deep network, we construct a transformer architecture in Figure 3. Each layer of the proposed architecture only consists of the MSSA operator and a skip connection. For language tasks, we additionally incorporate LayerNorm before the MSSA operator to improve performance. The complete architecture is built by stacking such layers, along with essential task-specific pre-processing and post-processing steps, such as positional encoding, token embedding, and final task-specific head to adapt to different applications.

**Remark 1.** *Generally speaking, the standard decoder-only transformer architecture is composed of the following key components (Brown et al., 2020b; Radford et al., 2019; Vaswani et al., 2017): (1) positional encoding, (2) multi-head QKV self-attention mechanisms, (3) feed-forward MLP networks, (4) layer normalization, and (5) residual connections. In contrast, our proposed transformer architecture adopts a streamlined design by incorporating several key simplications. Specifically, it employs shared-QKV subspace self-attention mechanisms, excludes MLP layers, and reduces the frequency of LayerNorm.*

**Differences from previous works on attention-only transformers.** In the literature, some theoretical works have studied attention-only transformers. For example, Dong et al. (2021); Wu et al. (2024) showed that pure-attention transformers with skip connections or LayerNorm can prevent rank collapse. Additionally, Alcalde et al. (2024) studied the clustering behavior of attention-only hardmax transformers. While these studies contribute significantly to our understanding of the role of self-attention in transformers, they lack empirical validation and practical implications. In contrast to these works, we not only show that each layer of the proposed attention-only transformer can denoise token representations but also conduct experiments on real-world language and vision tasks to demonstrate the potential.

**The role of backward propagation.** Notably, our approach constructs a transformer architecture in the forward pass by interpreting each layer as a denoising operator, conditioned on the assumption that the subspace bases $\{\boldsymbol{U}_k\}_{k=1}^K$ are known. However, in practice, these subspace matrices, i.e., network parameters, are unknown and need to be learned gradually via iterative optimization too. Hence, the forward denoising operator (3) at the $l$-th layer/iteration becomes

$$\boldsymbol{Z}^{(l+1)} = \boldsymbol{Z}^{(l)} + \eta \sum_{k=1}^{K} \boldsymbol{U}_k^{(l)} \boldsymbol{U}_k^{(l)^T} \boldsymbol{Z}^{(l)} \varphi\left(\boldsymbol{Z}^{(l)^T} \boldsymbol{U}_k^{(l)} \boldsymbol{U}_k^{(l)^T} \boldsymbol{Z}^{(l)}\right), \quad l = 0, 1, \ldots, L-1. \quad (5)$$

We should emphasize that the parameters $\{\boldsymbol{U}_k^{(l)}\}$ now depend on the layer index $l$ and can be different across layers. Note that $\boldsymbol{U}_k^{(l)}$ at different layers can represent different intermediate estimates for $\boldsymbol{U}_k$ via certain optimization. In practice, they can be estimated through end-to-end training via backpropagation. This flexibility brings additional capacity for the overall deep architecture, allowing learning denoising bases $\{\boldsymbol{U}_k^{(l)}\}$ at each layer that is locally adaptive to the distribution of $\boldsymbol{Z}^{(l)}$.


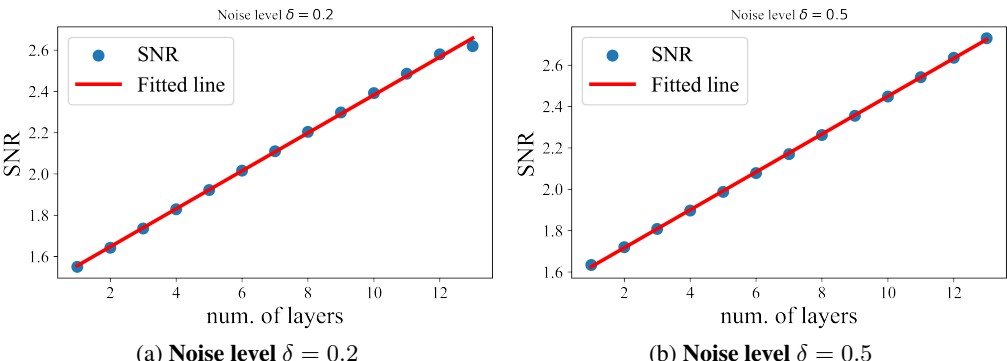

(a) **Noise level** $\delta = 0.2$          (b) **Noise level** $\delta = 0.5$

Figure 4: **Denosing performance of the attention-only transformer.** Here, we sample initial token representations from a mixture of low-rank Gassuains in Definition 1. Then, we apply (3) to update token representations and report the SNR at each layer.

## 3 THEORETICAL GUARANTEE FOR THE ATTENTION-ONLY TRANSFORMER

In this section, we rigorously show that each layer of the proposed transformer denoises token representation when the initial token representations are sampled from a mixture of low-rank Gaussians as defined in Definition 1. To quantify the denoising capability, we define the signal-to-noise ratio (SNR) for each block of the token representations at the $l$-th layer as follows:

$$\mathrm{SNR}(\boldsymbol{Z}_k^{(l)}) := \frac{\|\boldsymbol{U}_k\boldsymbol{U}_k^T\boldsymbol{Z}_k^{(l)}\|_F}{\|(\boldsymbol{I} - \boldsymbol{U}_k\boldsymbol{U}_k^T)\boldsymbol{Z}_k^{(l)}\|_F}, \quad \forall k \in [K]. \tag{6}$$

To simplify our analysis, we assume that $p = p_1 = \cdots = p_K$, $N_1 = \cdots = N_K = N/K$, and

$$[\boldsymbol{U}_1 \quad \dots \quad \boldsymbol{U}_K] \in \mathcal{O}^{d \times Kp}. \tag{7}$$

With the above setup, we now characterize the denoising performance of the proposed transformer.

**Theorem 1.** *Let $\boldsymbol{Z}^{(0)}$ be defined in Definition 1 and $\varphi(\cdot)$ in Eq. (4) be $\varphi(\boldsymbol{x}) = h(\sigma(\boldsymbol{x}))$, where $\sigma : \mathbb{R}^N \to \mathbb{R}^N$ is the soft-max function and $h : \mathbb{R}^N \to \mathbb{R}^N$ is an element-wise thresholding function with $h(x) = \tau\mathbb{I}\{x > \tau\}$ for each $i \in [N]$. Suppose that $p \gtrsim \log N$, $\delta \lesssim \sqrt{\log N}/\sqrt{p}$, and*

$$\tau \in \left(\frac{1}{2}, \frac{1}{1 + N\exp(-9p/32)}\right].$$

*For sufficiently large $N$, it holds with probability at least $1 - KLN^{-\Omega(1)}$ that for each $l \in [L-1]$,*

$$\mathrm{SNR}(\boldsymbol{Z}_k^{(l+1)}) = (1 + \eta\tau)\mathrm{SNR}(\boldsymbol{Z}_k^{(l)}), \quad \forall k \in [K]. \tag{8}$$

The proof is deferred to Appendix A. Here we comment on significance of this theorem:

- **Linear denoising performance of the attention-only transformer.** In the theorem, when the initial token representations are sampled from a mixture of low-rank Gaussian distributions with a noise level $O(\sqrt{\log N}/\sqrt{p})$ and $\varphi(\cdot)$ is defined in (4), we show that each layer of the proposed transformer denoises token representations at a linear rate. This indicates the MSSA operator's efficiency in reducing noise across layers. Notably, our theoretical results are well-supported by experimental observations in Figure 4, which further validate the practical denoising capability of the proposed transformer.

- **Difficulties in analyzing the dynamics of the update (3).** It is worth noting that the update (3) is highly nonlinear and complicated. Specifically, it is cubic in terms of update variables $\boldsymbol{Z}^{(l)}$ and the operator $\varphi$ is nonlinear, being composed of soft-max and thresholding functions. These characteristics lead to intricate interactions among consecutive updates that complicate the analysis of the learning dynamics. Compared to the existing works (Ahn et al., 2023; Zhang et al., 2023; Schlag et al., 2021) that mainly focus on linear self-attention with $\varphi(\cdot)$ being the identify function, our analysis provides more pertinent results for understanding the denoising performance and learning dynamics of attention mechanisms, capturing the *nonlinear* interactions and transformations across the layers of modern transformer architectures.

## 4 EXPERIMENTAL RESULTS

In this section, we evaluate our proposed *attention-only transformer* (AoT) architecture on both language and vision tasks. Due to limited computing and engineering resources, the goal of our experimentation is not to outperform state-of-the-art transformers but to verify that AoT can achieve similar or comparable performance on complex language and vision tasks. Hence we believe, while offering a fully interpretable architecture with a layerwise performance guarantee, AoT holds great potential in practical applicability with further engineering development in the future. In all our implementations, we set the operator $\varphi(\cdot)$ in Eq. (3) to be the softmax function.

### 4.1 DECODER-ONLY TRANSFORMER FOR LANGUAGE TASKS

To study the performance of our architecture on language tasks, we consider the widely used Generative Pre-Training (GPT) task (Radford et al., 2019). In the context of causal language modeling, the goal is to do the next token prediction in a sequence. To adapt to this task, we modify the AoT architecture by changing the MSSA operator to be a causally masked MSSA, i.e., replacing (5) by

$$\boldsymbol{Z}^{(l+1)} = \boldsymbol{Z}^{(l)} + \eta \sum_{k=1}^{K} \boldsymbol{U}_k^{(l)} \boldsymbol{U}_k^{(l)^T} \boldsymbol{Z}^{(l)} \varphi\left(\text{Mask}\left(\boldsymbol{Z}^{(l)^T} \boldsymbol{U}_k^{(l)} \mathcal{P}\left(\boldsymbol{U}_k^{(l)^T} \boldsymbol{Z}^{(l)}\right)\right)\right),$$

where $[\text{Mask}(\boldsymbol{A})]_{ij} = a_{ij}$ if $i \leq j$ and $[\text{Mask}(\boldsymbol{A})]_{ij} = -\infty$ otherwise. Following the implementation used in Kitaev et al. (2020), we apply normalization to the "query matrix" $\boldsymbol{U}_k^{(l)^T} \boldsymbol{Z}^{(l)}$, where $\boldsymbol{A}' = \mathcal{P}(\boldsymbol{A})$ project each column of $\boldsymbol{A} = [\boldsymbol{a}_1, \ldots, \boldsymbol{a}_n] \in \mathbb{R}^{d \times n}$ onto unit sphere, i.e., $\boldsymbol{a}' = \boldsymbol{a}/\|\boldsymbol{a}\|$. We follow the same pre-processing and post-processing steps in (Yu et al., 2024, Section 4.1.4). Our implementation of the GPT-2 type transformer and training pipeline is based on the framework outlined in Karpathy (2022).[3] In addition, to study the effect of removing the MLP layer, we also train models with MLPs in the first half of transformer blocks, referred as *Hybrid*, as well as models with MLPs in all blocks, referred as *Full MLP*.

#### 4.1.1 LANGUAGE MODELING

**Pre-training language models.** We pre-train AoT-based language models of different sizes and GPT-2 (see Table 1 for model sizes) on OpenWebText (Gokaslan & Cohen, 2019). Here, we train these models over a 1024-token context using the AdamW optimizer (Loshchilov & Hutter, 2019). We plot the training loss and validation loss against the number of training iterations in Figure 5(a) and (b), respectively. It is observed that AoT-based language models of medium and large size can achieve comparable performance to the GPT-2 base model in terms of training and validation loss. In addition, a comparison of AoT models with the Hybrid and Full MLP configurations demonstrates that incorporating MLP layers can accelerate the training process.

**Zero-shot evaluation.** Using the above pre-trained models, we compute the cross-entropy validation loss without training on datasets WikiText (Merity et al., 2016)[4], LAMBADA (Paperno et al., 2016)[5], and PTB (Marcus et al., 1993) in Table 1. In addition, we report zero-shot accuracy in Table 1 on LAMBADA for predicting the final word of sentences, as well as on the Children's Book Test (CBT) (Hill et al., 2016), where the task is to choose either common nouns (CN) or named entities (NE) from 10 possible options for an omitted word in a paragraph. It is observed that AoT with medium and large parameter sizes can achieve comparable performance to the GPT-2 base model on these tasks. Moreover, we found that adding MLP layers to AoT does not improve the zero-shot performance. These results highlight the potential of attention-only models to achieve competitive results while maintaining interpretability.

#### 4.1.2 IN-CONTEXT LEARNING ON SIMPLE FUNCTION CLASSES

In-context learning (ICL) refers to the ability of modern language models to perform tasks by using examples provided in the input prompt, along with a new query input, generating outputs without

---

[3]https://github.com/karpathy/nanoGPT.git

[4]For WikiText2 and WikiText103 (Merity et al., 2016), the test splits are the same, so we merge them as a single dataset referred to as WikiText.

[5]To obtain the accuracy on LAMBADA dataset, we use greedy decoding.

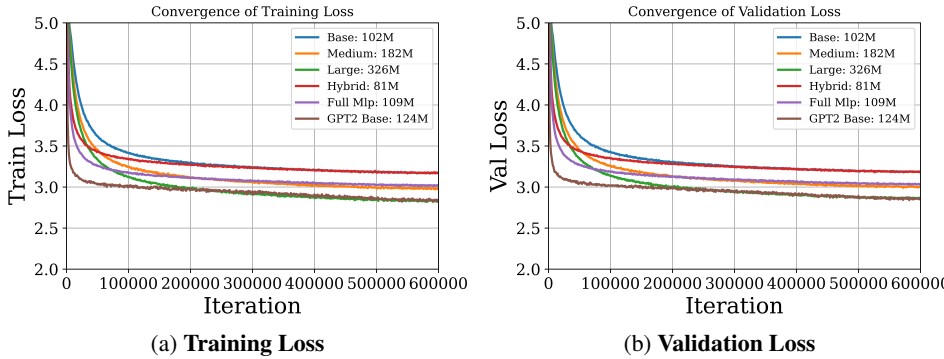

(a) **Training Loss**  (b) **Validation Loss**

Figure 5: The curves of both training and validation losses of models pretrained on OpenWebText.

updating the parameters (Brown et al., 2020b; Garg et al., 2023; Park et al., 2024). We evaluate the ICL capabilities of our AoT and compare its performance with that of GPT-2 (Radford et al., 2019). Each model is trained from scratch on specific tasks, including linear and sparse linear regressions. We mainly follow the setup in Garg et al. (2023) to train models to learn linear functions in context. Specifically, for a specific function class $\mathcal{G}$, we generate random prompts by sampling a function $g \in \mathcal{G}$ from distribution $\mathcal{D}_{\mathcal{G}}$ over functions random inputs $\boldsymbol{x}_1, \ldots, \boldsymbol{x}_N \in \mathbb{R}^d$ i.i.d. from $\mathcal{D}_{\mathcal{X}}$ over inputs. To evaluate the inputs on $g$, we create a prompt $P = (\boldsymbol{x}_1, g(\boldsymbol{x}_1), \ldots, \boldsymbol{x}_N, g(\boldsymbol{x}_N))$. We train the model $f_{\boldsymbol{\theta}}$ to minimize the expected loss over all prompts prefixes:

$$\min_{\boldsymbol{\theta}} \mathbb{E}_P \left[ \frac{1}{N} \sum_{i=1}^{N-1} \left( f_{\boldsymbol{\theta}}(P^i) - g(\boldsymbol{x}_i) \right)^2 \right], \tag{9}$$

where $P^i$ is the prompt prefix up to the input $i$-th in-context example $P = (\boldsymbol{x}_1, g(\boldsymbol{x}_1), \ldots, \boldsymbol{x}_i)$.

**Tasks.** We consider both linear functions and sparse linear functions with dimension $d = 20$. The in-context examples $\boldsymbol{x}_i$ are sampled from the isotropic Gaussian distribution. For linear functions, we define $\mathcal{G} = \{g : g(\boldsymbol{x}) = \boldsymbol{w}^T \boldsymbol{x}\}$, where $\boldsymbol{x}$ is sampled from the isotropic Gaussian distribution as well. For sparse linear functions, the setup is similar, but with a modification: only 3 coordinates in the vector $\boldsymbol{w}$ are set as non-zero, while the remaining coordinates are set as zero.

**Training and evaluation.** For all experiments, we set the number of heads to 8 and embedding size 128. To match the sizes of different models by controlling the number of layers. The transformer and Full MLP has 16 layers, Hybrid 24, and AoT 16. To train the model, we sample a batch of random prompts with size 64 and train the models for 50,000 iterations using Adam optimizer (Kingma & Ba, 2017) . We evaluate models using same $\mathcal{D}_{\mathcal{G}}$ and $\mathcal{D}_{\mathcal{X}}$ to sample 1280 prompts. We refer the reader to Park et al. (2024) for more details.

Table 1: Zero-shot results on several benchmark datasets.

| Models
# of parameters | LAMBADA
(val loss) ↓ | PTB
(val loss) ↓ | WikiText
(val loss) ↓ | LAMBADA
(acc) ↑ | CBT CN
(acc) ↑ | CBT NE
(acc) ↑ |
|---|---|---|---|---|---|---|
| Base 102M | 4.70 | 6.03 | 4.65 | 0.25 | 0.80 | 0.74 |
| Medium 182M | 4.47 | **5.08** | 4.22 | 0.29 | 0.84 | 0.77 |
| Large 326M | **4.26** | **4.77** | **3.99** | **0.34** | **0.86** | **0.81** |
| Hybrid 81M | 4.84 | 5.83 | 4.56 | 0.25 | 0.79 | 0.73 |
| Full MLP 109M | 4.73 | 6.95 | 4.70 | 0.30 | 0.83 | 0.77 |
| GPT-2 Base 124M | **4.32** | 5.75 | **4.13** | **0.40** | **0.87** | **0.84** |

We plot the estimation error against in-context samples in Figure 6. It is observed that our AoT architecture can in-context learn linear functions and sparse linear functions, achieving performance close to that of GPT-2 style transformer. Adding MLPs does not improve the in-context learning ability of AoT, which further supports the effectiveness of our attention-only architecture.

### 4.2 VISION TRANSFORMERS FOR SUPERVISED IMAGE CLASSIFICATION

Now we evaluate the performance of AoT as a backbone architecture for supervised image classification tasks. For further simplification, we do *not* even use LayerNorm layers in the AoT architecture.

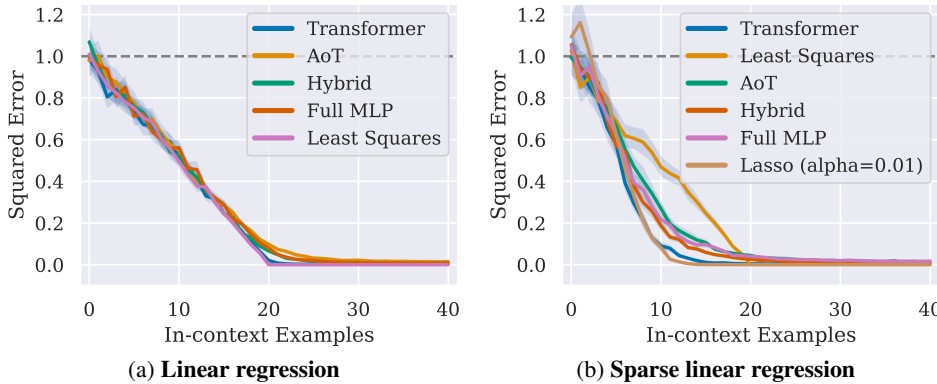

Figure 6: Evaluating models on in-context learning linear functions. We plot the normalized squared error as a function of in-context examples.

**Model architecture.** As we mentioned earlier, for vision tasks, we can use an even simpler architecture without the Layernorm (see Figure 3). We use the same pre-processing map and classification head defined in (Yu et al., 2023a, Section 4.1.1) to construct the AoT-based model. Moreover, we consider AoT-based models with different number of parameters and attention layers, as in Table 2.

Table 2: Top-1 accuracy on ImageNet with different models when pre-trained on ImageNet-21K and then fine-tuned on ImageNet-1K.

| Models | ImageNet-1K | # of Parameters | # of Layers |
|---|---|---|---|
| AoT-Base | 70.2% | 16M | 12 (`Atten`) |
| AoT-Large | 75.7% | 52M | 24 (`Atten`) |
| AoT-Huge | 79.2% | 86M | 32 (`Atten`) |
| CRATE-$\alpha$-B/16 (Yang et al., 2024) | 81.2% | 72.3M | 12 (`Atten+MLP`) |
| CRATE-$\alpha$-L/14 (Yang et al., 2024) | 83.9% | 253.8M | 24 (`Atten+MLP`) |

**Training setup.** We employ Lion optimizer (Chen et al., 2024) to pre-train the above AoT-based transformer on ImageNet-21K and AdamW (Loshchilov, 2017) to fine-tune it on ImageNet-1K (Deng et al., 2009) by minimizing the cross-entropy (CE) loss. During the pre-training, we set the learning rate as $1 \times 10^{-4}$, weight decay as $0.05$, and batch size as $4096$. During the fine-tuning, the learning rate as $5 \times 10^{-5}$, weight decay as $0.05$, and batch size as $2048$. Standard data augmentation techniques, including random cropping, random horizontal flipping, and random augmentation, are used in our implementation, the same as those used in Yu et al. (2023b).

Based on the above experimental setup, we report the top-1 accuracy of AoT on ImageNet-1K in Table 2. For comparison, we also report the performance of CRATE-$\alpha$ models in Yang et al. (2024), which are enhanced white-box vision models built on CRATE (Yu et al., 2023b). Despite the absence of MLP layers in AoT, it achieves a competitive performance comparable to that of CRATE. This result demonstrates the effectiveness and efficiency of the attention-only architecture.

## 5 CONCLUSION

In this work, we propose a new and minimalistic transformer architecture by interpreting each layer as the application of a subspace denoising operator to token representations, where these representations are assumed to be sampled from a mixture of low-rank Gaussians. Remarkably, this architecture consists of subspace self-attention layers and skip connections at each layer, without the MLP operators at all. We have shown that each such layer improves the signal-to-noise ratio of token representations at a linear rate with respect to the number of layers. We have verified the practical potential of this simple architecture through extensive experiments on both language and vision tasks, which strongly suggest that it could lead to more efficient and effective architectures in the future.

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

To simplify our development, we introduce some further notation. We use $\mathrm{BlkDiag}(\boldsymbol{X}_1, \ldots, \boldsymbol{X}_K)$ to denote a block diagonal matrix whose diagonal blocks are $\boldsymbol{X}_1, \ldots, \boldsymbol{X}_K$.

# A PROOF OF THEOREM 1

## A.1 PRELIMINARY RESULTS

To prove Theorem 1, we first establish several probabilistic results about Gaussian random vectors. First, we present a probabilistic bound on the deviation of the norm of Gaussian random vectors from its mean. This is an extension of (Vershynin, 2018, Theorem 3.1.1).

**Lemma 1.** *Let $\boldsymbol{x} \sim \mathcal{N}(\boldsymbol{0}, \delta^2 \boldsymbol{I}_d)$ be a Gaussian random vector. It holds with probability at least $1 - 2\exp\left(-t^2/2\delta^2\right)$ that*

$$\left| \|\boldsymbol{x}\| - \delta\sqrt{d} \right| \leq t + 2\delta. \tag{10}$$

Based on the above lemma, we can respectively estimate the norm of coefficients in the signal and noise parts, the products between different pairs of Gaussian random vectors, and the bounds on the soft-max values of these products.

**Lemma 2.** *Consider the setting in Definition 1 with $p = p_1 = \cdots = p_K$ and $N_1 = \cdots = N_K = N/K$. Suppose that $p \geq 16(\sqrt{\log N} + 1)^2$ and*

$$N \geq 8\pi K^2 \log^3 N, \; \delta \leq \frac{1}{8}\sqrt{\frac{\log N}{p}}. \tag{11}$$

*The following statements hold:*
*(i) With probability at least $1 - 2KN^{-1}$, we have*

$$\left| \|\boldsymbol{a}_i\| - \sqrt{p} \right| \leq 2\left(\sqrt{\log N} + 1\right), \forall i \in [N], \tag{12}$$

$$\left| \|\boldsymbol{e}_{i,l}\| - \delta\sqrt{p} \right| \leq 2\delta\left(\sqrt{\log N} + 1\right), \forall i \in C_k, l \neq k \in [K]. \tag{13}$$

*(ii) With probability at least $1 - 4KN^{-2}$, we have*

$$|\langle \boldsymbol{a}_i, \boldsymbol{a}_j \rangle| \leq 3\sqrt{\log N} \|\boldsymbol{a}_i\|, \forall i \neq j \in C_k, k \in [K], \tag{14}$$

$$|\langle \boldsymbol{a}_i, \boldsymbol{e}_{j,l} \rangle| \leq 3\sqrt{\log N} \|\boldsymbol{e}_{j,l}\|, \forall i \in C_k, j \in C_l, k \neq l \in [K], \tag{15}$$

$$|\langle \boldsymbol{e}_{i,k}, \boldsymbol{e}_{j,k} \rangle| \leq 3\delta\sqrt{\log N} \|\boldsymbol{e}_{j,k}\|, \forall i \in C_l, j \in C_m, l, m \neq k. \tag{16}$$

*(iii) With probability at least $1 - 2N^{-1}$, we have*

$$\max_{i \in C_k} \langle \boldsymbol{a}_i, \boldsymbol{e}_{j,k} \rangle \geq \sqrt{\log N} \|\boldsymbol{e}_{j,k}\|, \forall j \in C_l, l \neq k \in [K]. \tag{17}$$

*(iv) With probability at least $1 - 4KN^{-1}$, we have*

$$\frac{\exp\left(\langle \boldsymbol{a}_i, \boldsymbol{e}_{j,k} \rangle\right)}{\sum_{i' \in C_k} \exp\left(\langle \boldsymbol{a}_{i'}, \boldsymbol{e}_{j,k} \rangle\right)} \leq \frac{1}{2}, \forall i \in C_k, j \in C_l, k \neq l \in [K], \tag{18}$$

$$\frac{\exp\left(\langle \boldsymbol{e}_{i,k}, \boldsymbol{e}_{j,k} \rangle\right)}{\sum_{i' \neq j, i' \in C_l} \exp\left(\langle \boldsymbol{e}_{i',k}, \boldsymbol{e}_{j,k} \rangle\right)} \leq \frac{1}{2}, \forall i \neq j, i \in C_l, j \in C_m, l, m \neq k. \tag{19}$$

*Proof.* (i) Applying Lemma 1 to $\boldsymbol{a}_i \sim \mathcal{N}(\boldsymbol{0}, \boldsymbol{I}_p)$ with $t = 2\sqrt{\log N}$ yields

$$\mathbb{P}\left( \left| \|\boldsymbol{a}_i\| - \sqrt{p} \right| \leq 2(\sqrt{\log N} + 1) \right) \geq 1 - 2N^{-2}.$$

This, together with the union bound, yields that (12) holds for all $i \in [N]$ with probability at least $1 - 2N^{-1}$. Using the same argument, we obtain that (13) holds for all $i \in C_k$ and $l \neq k \in [K]$ with probability at least $1 - 2(K - 1)N^{-1}$. Finally, applying the union bound yields that the probability is $1 - 2KN^{-1}$.

(ii) For each pair $(i, j)$ with $i \neq j \in C_k$ and $k \in [K]$, conditioned on $\boldsymbol{a}_i$, we have $\langle \boldsymbol{a}_i, \boldsymbol{a}_j \rangle \sim \mathcal{N}(0, \|\boldsymbol{a}_i\|^2)$. According to the tail bound the Gaussian random variable, we have

$$\mathbb{P}\left( |\langle \boldsymbol{a}_i, \boldsymbol{a}_j \rangle| \geq 3\|\boldsymbol{a}_i\|\sqrt{\log N} \Big| \boldsymbol{a}_i \right) \leq 2N^{-4}.$$

This, together with the union bound, implies that conditioned on $\boldsymbol{a}_i$, it holds with probability at least $1 - 2N^{-2}$ that $|\langle \boldsymbol{a}_i, \boldsymbol{a}_j \rangle| \leq 2\|\boldsymbol{a}_i\|\sqrt{\log N}$ for all $i \neq j \in C_k$ and $k \in [K]$. Using the same argument, we obtain (15) and (16). Finally, applying the union bound yields the probability.

(iii) Conditioned on $\boldsymbol{e}_{j,k}$, we obtain that $X_i := \langle \boldsymbol{a}_i, \boldsymbol{e}_{j,k} \rangle / \|\boldsymbol{e}_{j,k}\| \sim \mathcal{N}(0, 1)$ for each $i \in C_k$ are i.i.d. standard normal random variables. Then, we have

$$\mathbb{P}\left( \max_{i \in C_k} X_i \geq \sqrt{\log N} \right) = 1 - \left( \mathbb{P}\left( X_1 < \sqrt{\log N} \right) \right)^{N_k}. \tag{20}$$

Using the property of the standard Gaussian random variable, we have

$$\mathbb{P}(X_1 \geq t) \geq \left( \frac{1}{t} - \frac{1}{t^3} \right) \frac{1}{\sqrt{2\pi}} \exp\left( -\frac{t^2}{2} \right).$$

Taking $t = \sqrt{\log N}$, we obtain

$$\mathbb{P}\left( X_1 \geq \sqrt{\log N} \right) = \frac{1}{\sqrt{\log N}} \left( 1 - \frac{1}{\log N} \right) \frac{1}{\sqrt{2\pi}} \exp\left( -\frac{\log N}{2} \right) \geq \frac{1}{2\sqrt{2\pi N \log N}}, \tag{21}$$

where the inequality follows from $N \geq \exp(2)$. Substituting this into (20) yields

$$\mathbb{P}\left( \max_{i \in C_k} X_i \geq \sqrt{\log N} \right) \geq 1 - \left( 1 - \frac{1}{2\sqrt{2\pi N \log N}} \right)^{N/K}$$

$$\geq 1 - \exp\left( -\frac{\sqrt{N}}{2K\sqrt{2\pi \log N}} \right) \geq 1 - N^{-1},$$

where the second inequality uses $1 - x \leq \exp(-x)$ for all $x > 0$ and the last inequality follows from $N \geq 8\pi K^2 \log^3 N$. This, together with the definition of $X_i$, implies (17).

(iv) Conditioned on $\boldsymbol{e}_{j,k}$, we have $X_i := \langle \boldsymbol{a}_i, \boldsymbol{e}_{j,k} \rangle \sim \mathcal{N}(0, \|\boldsymbol{e}_{j,k}\|^2)$ for each $i \in C_k$ are i.i.d. normal random variables. Suppose that (13) holds for all $i \in C_k, l \neq k \in [K]$, which happens with probability at least $1 - 2(K-1)N^{-1}$ according to (i). This implies for all $j \in C_k$ and $k \in [K]$,

$$\|\boldsymbol{e}_{j,k}\| \leq \delta\left( \sqrt{p} + 2\sqrt{\log N} + 2 \right) \leq \frac{3}{2}\delta\sqrt{p}, \tag{22}$$

where the last inequality follows from $p \geq 16(\sqrt{\log N} + 1)^2$ due to (11). For ease of exposition, let

$$\sigma := \|\boldsymbol{e}_{j,k}\|, \ S := \sum_{i \in C_k} \exp(X_i). \tag{23}$$

Obviously, showing (18) is equivalent to proving

$$2\exp(X_i) \leq \sum_{i' \in C_k} \exp(X_{i'}) = S, \ \forall i \in C_k. \tag{24}$$

Note that $X_i / \sigma \sim \mathcal{N}(0, 1)$ for all $i \in C_k$. Using the tail bound of the standard normal random variable, we have

$$\mathbb{P}\left( \frac{|X_i|}{\sigma} \geq 2\sqrt{\log N} \right) \leq 2N^{-2}, \ \forall i \in C_k.$$

This, together with the union bound, yields that it holds with probability $1 - 2N^{-1}$ that $|X_i| \leq 2\sigma\sqrt{\log N}$ for all $i \in [N]$. Using this, (22), (23), and the union bound, we obtain with probability at least $1 - 2KN^{-1}$,

$$|X_i| \leq 3\delta\sqrt{p \log N}, \ \forall i \in [N].$$

Therefore, we have

$$\exp\left(-3\delta\sqrt{p \log N}\right) \leq \exp(X_i) \leq \exp\left(3\delta\sqrt{p \log N}\right), \; \forall i \in [N]. \tag{25}$$

Using this and (23), we have

$$S \geq \frac{N}{K} \exp\left(-3\delta\sqrt{p \log N}\right).$$

This, together with (25), implies that proving (24) is sufficient to proving

$$\log N \geq 6\delta\sqrt{p \log N} + \log\left(2K\right),$$

which holds when $N \geq \max\{16K^4, \exp\left(64\delta^2 p\right)\}$ due to (11). According to the union bound, (18) holds with probability at least $1 - 2KN^{-1}$. Using the same argument, (19) holds with probability at least $1 - 2KN^{-1}$. $\qquad\square$

### A.2 Proof of Theorem 1

To simplify our development, let

$$M_1 := \begin{bmatrix} \theta^2 A_1^T A_1 & \theta A_1^T E_{2,1} & \dots & \theta A_1^T E_{K,1} \\ \theta E_{2,1}^T A_1 & E_{2,1}^T E_{2,1} & \dots & E_{2,1}^T E_{K,1} \\ \vdots & \vdots & \ddots & \vdots \\ \theta E_{K,1}^T A_1 & E_{K,1}^T E_{2,1} & \dots & E_{K,1}^T E_{K,1} \end{bmatrix} \in \mathbb{R}^{N \times N}, \tag{26}$$

$$M_2 := \begin{bmatrix} E_{1,2}^T E_{1,2} & \theta E_{1,2}^T A_2 & \dots & E_{1,2}^T E_{K,2} \\ \theta A_2^T E_{1,2}^T & \theta^2 A_2^T A_2 & \dots & \theta A_2^T E_{K,2} \\ \vdots & \vdots & \ddots & \vdots \\ E_{K,2}^T E_{1,2} & \theta E_{K,2}^T A_2 & \dots & E_{K,2}^T E_{K,2} \end{bmatrix} \in \mathbb{R}^{N \times N},$$

$$\vdots$$

$$M_K := \begin{bmatrix} E_{1,K}^T E_{1,K} & E_{1,K}^T E_{2,K} & \dots & \theta E_{1,K}^T A_K \\ E_{2,K}^T E_{1,K} & E_{2,K}^T E_{2,K} & \dots & \theta E_{2,K}^T A_k \\ \vdots & \vdots & \ddots & \vdots \\ \theta A_K^T E_{1,K} & \theta A_K^T E_{2,K} & \dots & \theta^2 A_K^T A_K \end{bmatrix} \in \mathbb{R}^{N \times N}.$$

where $\theta \geq 1$. Recall that

$$Z^{(0)} = \begin{bmatrix} Z_1^{(0)} & \dots & Z_K^{(0)} \end{bmatrix} = \begin{bmatrix} U_1 A_1 + \sum_{j \neq 1} U_j E_{1,j} & \dots & U_K A_K + \sum_{j \neq K} U_j E_{K,j} \end{bmatrix}, \tag{27}$$

**Lemma 3.** *Consider the setting in Definition 1 with $p = p_1 = \dots = p_K$ and $N_1 = \dots = N_K = N/K$. Let $\varphi(\cdot)$ be*

$$\varphi(x) = h(\sigma(x)), \tag{28}$$

*where $\sigma : \mathbb{R}^N \to \mathbb{R}^N$ is the soft-max function and $h : \mathbb{R}^N \to \mathbb{R}^N$ is an element-wise thresholding function with $h(x) = \tau \mathbb{I}\{x > \tau\}$ for each $i \in [N]$. Suppose that (11) holds. Suppose in addition that $p \geq 64(\sqrt{\log N} + 1)^2$ and*

$$\tau \in \left(\frac{1}{2}, \frac{1}{1 + N\exp(-9p/32)}\right] \tag{29}$$

*The following statements hold with probability at least $1 - KN^{-\Omega(1)}$ that ,*

$$\varphi(M_1) = \text{BlkDiag}(\tau I, 0, \dots, 0), \; \dots, \; \varphi(M_K) = \text{BlkDiag}(0, 0, \dots, \tau I). \tag{30}$$

*Proof.* Suppose that (12)-(19) hold, which happens with probability at least $1 - KN^{-\Omega(1)}$ according to Lemma 2, (11), and the union bound. Now, we focus on studying $\boldsymbol{M}_1$ as defined in (26). For ease of exposition, we denote the $i$-th column of $\boldsymbol{M}_1$ by $\boldsymbol{m}_i \in \mathbb{R}^N$ for each $i \in [N]$. Moreover, recall that

$$C_1 = \left\{1, 2, \ldots, \frac{N}{K}\right\}, \ldots, C_K = \left\{\frac{(K-1)N}{K} + 1, \frac{(K-1)N}{K} + 2, \ldots, N\right\}.$$

We now divide our proof into two cases. We first study the $i$-th column of $\boldsymbol{M}_1$ for each $i \in C_1$, and then study the $i$-th column of $\boldsymbol{M}_1$ for each $i \in C_k$ with $k \neq 1$.

**Case 1.** According to (26), we have for each $i \in C_1$,

$$m_{ij} = \theta^2 \langle \boldsymbol{a}_i, \boldsymbol{a}_j \rangle, \forall j \in C_1, \ m_{ij} = \theta \langle \boldsymbol{a}_i, \boldsymbol{e}_{j,k} \rangle, \forall j \in C_k, k \neq 1.$$

For each pair $(i, j)$ with $i \neq j \in C_1$, we compute

$$\frac{\sigma_i(\boldsymbol{m}_i)}{\sigma_j(\boldsymbol{m}_i)} = \exp\left(m_{ii} - m_{ij}\right) \geq \exp\left(\theta\|\boldsymbol{a}_i\|\left(\theta\|\boldsymbol{a}_i\| - 3\sqrt{\log N}\right)\right) \geq \exp\left(\frac{9\theta^2 p}{32}\right), \quad (31)$$

where the first inequality follows from (14) and the second uses (12) and $\sqrt{p} \geq 8(\sqrt{\log N} + 1)$. Using the same argument, for each pair $(i, j)$ with $i \in C_1$, $j \in C_k$, and $k \neq 1$, we obtain

$$\frac{\sigma_i(\boldsymbol{m}_i)}{\sigma_j(\boldsymbol{m}_i)} \geq \exp\left(\frac{9\theta^2 p}{32}\right),$$

This, together with $\sum_{j=1}^N \sigma_j(\boldsymbol{m}_i) = 1$, yields $\left(1 + (N-1)\exp\left(-9\theta^2 p/32\right)\right)\sigma_i(\boldsymbol{m}_i) \geq 1$. Therefore, we have for each $i \in C_1$,

$$\sigma_i(\boldsymbol{m}_i) \geq \frac{1}{1 + N\exp(-9\theta^2 p/32)} > \frac{1}{2}, \ \sigma_j(\boldsymbol{m}_i) \leq \frac{1}{2}, \ \forall j \neq i, \quad (32)$$

where the last inequality follows from $p \geq 64(\sqrt{\log N} + 1)^2$. This, together with the value of $\tau$ in (29), yields for each $i \in C_1$,

$$\sigma_j(\boldsymbol{m}_i) < \tau < \sigma_i(\boldsymbol{m}_i), \ \forall j \neq i.$$

Using this and (28), we have for each $i \in C_1$,

$$h\left(\sigma_i(\boldsymbol{m}_i)\right) = \tau, \ h\left(\sigma_j(\boldsymbol{m}_i)\right) = 0, \ \forall j \neq i.$$

**Case 2.** For each $i \in C_k$ with $k \neq 1$, it follows from (26) that

$$m_{ij} = \theta \langle \boldsymbol{e}_{i,1}, \boldsymbol{a}_j \rangle, \forall j \in C_1, \ m_{ij} = \langle \boldsymbol{e}_{i,1}, \boldsymbol{e}_{j,1} \rangle, \ \forall j \in C_l, l \neq 1.$$

Consider a fixed $i \in C_k$ with $k \neq 1$, it follows from (17) that there exists $j_i \in C_1$ such that $m_{ij_i} \geq \theta\|\boldsymbol{e}_{i,1}\|\sqrt{\log N}$. This implies

$$\frac{\sigma_{j_i}(\boldsymbol{m}_i)}{\sigma_i(\boldsymbol{m}_i)} = \exp\left(\theta m_{ij_i} - m_{ii}\right) \geq \exp\left(\|\boldsymbol{e}_{i,1}\|\left(\theta\sqrt{\log N} - \|\boldsymbol{e}_{i,1}\|\right)\right)$$

$$\geq \exp\left(\frac{3\delta\theta}{4}\sqrt{p\log N} - \frac{25}{16}\delta^2 p\right),$$

where the second inequality follows from (13). This, together with $\sigma_i(\boldsymbol{m}_i) + \sigma_{j_i}(\boldsymbol{m}_i) < 1$, implies

$$\sigma_i(\boldsymbol{m}_i) < \frac{1}{1 + \exp\left(3\delta\theta\sqrt{p\log N}/4 - 25\delta^2 p/16\right)} < \frac{1}{1 + \exp\left(\delta\theta\sqrt{p\log N}/2\right)} < \frac{1}{2}, \quad (33)$$

where the second inequality uses $\delta\sqrt{p} \leq \sqrt{\log N}/8$ due to (11). On the other hand, it follows from (18) and (19) that

$$\sigma_j(\boldsymbol{m}_i) \leq \frac{1}{2}, \forall j \neq i.$$

This, together with (33), $\delta \leq 1/8$, $\sqrt{p} \geq 8(\sqrt{\log N} + 1)$, and the value of $\tau$ by (29), yields for each $i \in C_k$ with $k \neq 1$,

$$\sigma_j(\boldsymbol{m}_i) < \tau, \forall j \in [N]. \tag{34}$$

This directly implies

$$h\left(\sigma(\boldsymbol{m}_i)\right) = \boldsymbol{0}, \ \forall i \in C_k, k \neq 1.$$

Then, we have $\varphi(\boldsymbol{M}_1) = \begin{bmatrix} \tau \boldsymbol{I} & \boldsymbol{0} \\ \boldsymbol{0} & \boldsymbol{0} \end{bmatrix}$. Applying the same argument to $\boldsymbol{M}_2, \ldots, \boldsymbol{M}_K$, we obtain (30). $\hspace{1em}\square$

Armed with the above result, we are ready to prove Theorem 1.

*Proof of Theorem 1.* For ease of exposition, let $\boldsymbol{M}_k^{(l)} := \boldsymbol{Z}^{(l)^T} \boldsymbol{U}_k \boldsymbol{U}_k^T \boldsymbol{Z}^{(l)}$ for each $k \in [K]$ and $l \in [L]$. Suppose that (30) holds, which happens with probability at least $1 - KN^{-\Omega(1)}$ according to (11), and (29), Lemma 3. We claim that for each $l \in [L]$, we have

$$\boldsymbol{Z}^{(l)} = \left[ (1 + \eta\tau)^l \boldsymbol{U}_1 \boldsymbol{A}_1 + \sum_{j \neq 1} \boldsymbol{U}_j \boldsymbol{E}_{1,j} \quad \ldots \quad (1 + \eta\tau)^l \boldsymbol{U}_K \boldsymbol{A}_K + \sum_{j \neq K} \boldsymbol{U}_j \boldsymbol{E}_{K,j} \right]. \tag{35}$$

This, together with (6), yields for each $k \in [K]$ and $l \in [L]$,

$$\mathrm{SNR}(\boldsymbol{Z}_k^{(l)}) = \frac{\|\boldsymbol{U}_k \boldsymbol{U}_k^T \boldsymbol{Z}_k^{(l)}\|_F}{\|(\boldsymbol{I} - \boldsymbol{U}_k \boldsymbol{U}_k^T)\boldsymbol{Z}_k^{(l)}\|_F} = \frac{(1 + \eta\tau)^l \|\boldsymbol{A}_k\|_F}{\|\sum_{j \neq k} \boldsymbol{U}_j \boldsymbol{E}_{k,j}\|_F},$$

which directly implies (8) for each $k \in [K]$ and $l \in [L-1]$. According to the union bound, the probability is $1 - KLN^{-\Omega(1)}$.

The rest of the proof is devoted to proving the claim (35) using the induction method. First, we consider the base case $l = 1$. According to (27) and (7), we compute

$$\boldsymbol{U}_1 \boldsymbol{U}_1^T \boldsymbol{Z}^{(0)} = [\boldsymbol{U}_1 \boldsymbol{A}_1 \quad \boldsymbol{U}_1 \boldsymbol{E}_{2,1} \quad \ldots \quad \boldsymbol{U}_1 \boldsymbol{E}_{K,1}],$$

$$\boldsymbol{M}_1^{(0)} = (\boldsymbol{U}_1 \boldsymbol{U}_1^T \boldsymbol{Z}^{(0)})^T(\boldsymbol{U}_1 \boldsymbol{U}_1^T \boldsymbol{Z}^{(0)}) = \begin{bmatrix} \boldsymbol{A}_1^T \boldsymbol{A}_1 & \boldsymbol{A}_1^T \boldsymbol{E}_{2,1} & \ldots & \boldsymbol{A}_1^T \boldsymbol{E}_{K,1} \\ \boldsymbol{E}_{2,1}^T \boldsymbol{A}_1 & \boldsymbol{E}_{2,1}^T \boldsymbol{E}_{2,1} & \ldots & \boldsymbol{E}_{2,1}^T \boldsymbol{E}_{K,1} \\ \vdots & \vdots & \ddots & \vdots \\ \boldsymbol{E}_{K,1}^T \boldsymbol{A}_1 & \boldsymbol{E}_{K,1}^T \boldsymbol{E}_{2,1} & \ldots & \boldsymbol{E}_{K,1}^T \boldsymbol{E}_{K,1} \end{bmatrix}.$$

Using the same argument, we can compute $\boldsymbol{M}_k^{(0)}$ for each $k \in [K]$. This, together with (30) for each $k \in [K]$, yields

$$\sum_{k=1}^K \boldsymbol{U}_k \boldsymbol{U}_k^T \boldsymbol{Z}^{(0)} \varphi(\boldsymbol{M}_k^{(0)}) = [\tau \boldsymbol{U}_1 \boldsymbol{A}_1 \quad \tau \boldsymbol{U}_2 \boldsymbol{A}_2 \quad \ldots \quad \tau \boldsymbol{U}_K \boldsymbol{A}_K].$$

Using this, (27), and (3), we directly obtain that (35) holds for $l = 1$. Next, we consider the case $l \geq 2$. Suppose that (35) holds for some $l \geq 1$. We compute

$$\boldsymbol{U}_1 \boldsymbol{U}_1^T \boldsymbol{Z}^{(l)} = \left[ (1 + \eta\tau)^l \boldsymbol{U}_1 \boldsymbol{A}_1 \quad \boldsymbol{U}_1 \boldsymbol{E}_{2,1} \quad \ldots \quad \boldsymbol{U}_1 \boldsymbol{E}_{K,1} \right],$$

$$\boldsymbol{M}_1^{(l)} = \begin{bmatrix} (1 + \eta\tau)^{2l} \boldsymbol{A}_1^T \boldsymbol{A}_1 & (1 + \eta\tau)^l \boldsymbol{A}_1^T \boldsymbol{E}_{2,1} & \ldots & (1 + \eta\tau)^l \boldsymbol{A}_1^T \boldsymbol{E}_{K,1} \\ (1 + \eta\tau)^l \boldsymbol{E}_{2,1}^T \boldsymbol{A}_1 & \boldsymbol{E}_{2,1}^T \boldsymbol{E}_{2,1} & \ldots & \boldsymbol{E}_{2,1}^T \boldsymbol{E}_{K,1} \\ \vdots & \vdots & \ddots & \vdots \\ (1 + \eta\tau)^l \boldsymbol{E}_{K,1}^T \boldsymbol{A}_1 & \boldsymbol{E}_{K,1}^T \boldsymbol{E}_{2,1} & \ldots & \boldsymbol{E}_{K,1}^T \boldsymbol{E}_{K,1} \end{bmatrix}.$$

Using the same argument, we can compute $\boldsymbol{M}_k^{(l)}$ for each $k \in [K]$. This, together with (30) for each $k \in [K]$, yields

$$\sum_{k=1}^K \boldsymbol{U}_k \boldsymbol{U}_k^T \boldsymbol{Z}^{(0)} \varphi(\boldsymbol{M}_k^{(0)}) = \left[ (1 + \eta\tau)^l \tau \boldsymbol{U}_1 \boldsymbol{A}_1 \quad (1 + \eta\tau)^l \tau \boldsymbol{U}_2 \boldsymbol{A}_2 \quad \ldots \quad (1 + \eta\tau)^l \tau \boldsymbol{U}_K \boldsymbol{A}_K \right].$$

Using this, (27), and (3), we directly obtain that (35) holds for $l + 1$. Then, we prove the claim. $\hspace{1em}\square$

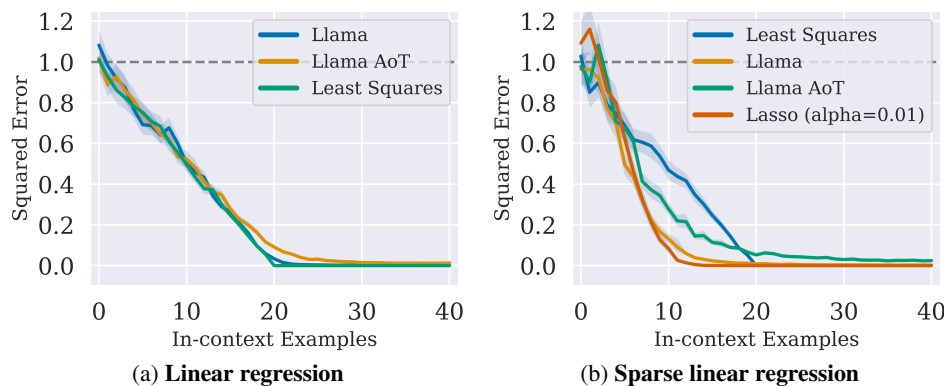

(a) **Linear regression**  (b) **Sparse linear regression**

Figure 7: Evaluating models of Llama architectures on in-context learning linear functions. We plot the normalized squared error as a function of in-context examples.

## B  SUPPLEMENTARY EXPERIMENTS

### B.0.1  MORE ON ICL

In addition, we performed the same ICL analysis as in Section 4.1.2. All the settings are the same, except that we changed the base model architecture to Llama (Touvron et al., 2023). And, we can see that the results are similar.

### B.0.2  EMERGENCE OF SEMANTIC PROPERTIES

The attention heads in our models have different semantic meanings, and indeed demonstrate the interpretability of our proposed architecture in practice. In Figure 8, we visualize the self-attention heatmaps between the [CLS] token and other image patches. We select 5 attention heads by manual inspection and find that they capture different parts of objects, displaying different semantic meanings.

### B.0.3  COMPUTING REQUIREMENT

In this section, we present the computing resources of a forward pass used by AoT-based language models and GPT-2 empirically in Table 3. The context window is 1024 tokens and the batch size is 16. The GFLOPS is measured by the PyTorch profiler, the total GPU memory consumption by the NVIDIA System Management Interface, and the running time of one forward pass by the Python time module. The only optimization we use is the default mode of the PyTorch compiler.

Table 3: The GFLOPS, total GPU memory consumption, and the running time of one forward pass are shown of AoT and GPT-2 at different sizes.

| Models | GFLOPS | Total GPU Memory in MiB | Running time in ms |
|---|---|---|---|
| Base 102M | 1651 | 21482 | 43 |
| Medium 182M | 3868 | 36198 | 78 |
| Large 326M | 8056 | 57896 | 225 |
| GPT-2 Base 124M | 2785 | 23300 | 27 |
| GPT-2 Medium 335M | 9898 | 51578 | 158 |

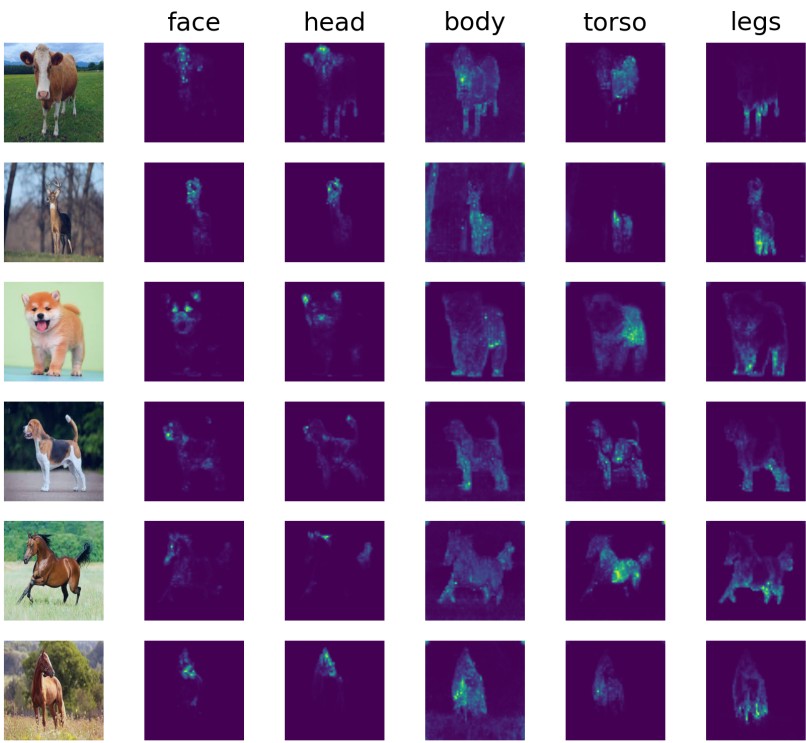

Figure 8: **Visualization of attention heads.** We feed our AoT a mini-batch of images and extract the attention maps of different heads from the penultimate layer. We show that these heads capture certain semantic meanings across different images.

