# OpenReview forum: "Attention-Only Transformers via Unrolled Subspace Denoising"
_ICLR.cc/2025/Conference — Submitted to ICLR 2025_

### Official Review · Reviewer_T5de · 2024-11-02

**Soundness:** 3
**Presentation:** 3
**Contribution:** 3
**Rating:** 6
**Confidence:** 2

**Summary:**

This paper proposes a multi-subspace self-attention (MSSA) to replace Transformers. By unrolling such iterative denoising operations as a deep network, the proposed technique arrives at a highly compact architecture that consists of only an MSSA operator with skip connections at each layer, without MLP. They also rigorously prove that each layer of the proposed transformer performs so highly efficient denoising that it improves the signal-to-noise ratio of token representations at a linear rate with respect to the number of layers.

**Strengths:**

1. The motivation is based on the existing linear representation hypothesis and the superposition hypothesis, which states that the token representations lie on a union of (possibly many) low-dimensional subspaces.

2. The empirical performance is good. On zero-shot language tasks and fine-tuning ViT, the method achieves comparable results.

**Weaknesses:**

1. A lot of linear hypothesis works can be used to do some interpretability tests of the LMMs. I am wondering if this new architecture might naturally fit the interpretability tests. I would suggest that the authors try GPT's auto-interpretability and might do some circuit analysis as done in [1].

2. Can the author do some experiments about training ViTs from scratch? That would be more convincing to show the effectiveness.

3. Can authors visualize some of the attention masks in ViTs for a few input images? That might help if each $U$ can correspond to different regions

>[1] Sparse Autoencoders Find Highly Interpretable Features in Language Models. ICLR 2024.

**Questions:**

Please see the weaknesses.

My main interest is to see if this architecture has better interpretability. The authors emphasize the interpretability in the intro but do not really validate it in the experiments. I am curious to see the results.

---

> ### Author Response · Authors · 2024-11-24
>
> **Q1.** A lot of linear hypothesis works can be used to do some interpretability tests of the LMMs. I am wondering if this new architecture might naturally fit the interpretability tests. I would suggest that the authors try GPT's auto-interpretability and might do some circuit analysis as done in [1].
>
> **A1.** Thanks for pointing this out. We agree that the proposed architecture naturally aligns with interpretability tests, given its minimalistic and theoretically grounded design. As recommended, we will explore GPT’s auto-interpretability framework and conduct circuit analysis similar to the approach in [1]. Due to current limitations in computational resources, we are working to implement these analyses and aim to update the results before the rebuttal deadline. We plan to incorporate these findings into the revised manuscript to provide additional insights into the interpretability of the proposed architecture.
>
> **Q2.**  Can the author do some experiments about training ViTs from scratch? That would be more convincing to show the effectiveness.
>
> **A2.**  Yes, we are running the experiment and plan to add them to our revised manuscript.
>
> **Q3.** Can authors visualize some of the attention masks in ViTs for a few input images? That might help if each U can correspond to different regions.
>
> **A3.** Yes. We have trained the proposed AoT architecture on Imagenet21k and visualized the self-attention heatmaps between the classification token and other image patches. Then we refer the reviewer to Figure 8 on Page 20 in the updated manuscript for these experimental results. It is observed that different heads capture different parts of objects, and display different semantic meanings.
>
> **Q4.** My main interest is to see if this architecture has better interpretability. The authors emphasize the interpretability in the intro but do not really validate it in the experiments. I am curious to see the results.
>
> **A4.** We should clarify that **interpretability in our work refers to understanding how the transformer architecture transforms token representations across layers**. This notion differs from the traditional definition of interpretability in large language models (LLMs), which often emphasizes extracting concise, human-readable explanations for specific token representations. Our approach to interpretability is rooted in the mathematical and structural understanding of the transformer’s behavior, providing a clearer view of how information is processed and denoised at each layer.

---

> > ### Comment · Reviewer_T5de · 2024-11-26
> > **Thanks for the response**
> >
> > Thanks for the response!
> >
> > I am curious to see the results of auto-interpretability and especially training ViTs from scratch. Fine-tuning ViTs pre-trained on ImageNet-21k is a bit tricky because the large-scale pre-training somehow gives guarantees about the performance. Training ViTs from scratch is more challenging and can show the efficacy of the proposed methods.
> >
> > For now, I will keep the original score unless some impressive updates are presented.

---

> > > ### Author Response · Authors · 2024-11-29
> > >
> > > Due to time and computational constraints, we first trained our AoT model on ImageNet-1k from scratch. The learning rate is 5e-4, weight decay 0.5 with Lion optimizer, and batch size 2048. We applied only basic augmentations, including random resized cropping and random horizontal flips, and did not extensively tune the hyperparameters. Below, we report the performance of our AoT model compared to the ViT Small:
> > >
> > > | Model | # of Parameters | Acc |
> > > | -------- | -------- | -------- |
> > > | AoT     |  15M    | 69.5%     |
> > > | ViT Small     | 22M     | 72.4%     |
> > >
> > > It is observed that the performance gap is relatively small, especially considering that our AoT model has significantly fewer parameters.  Moreover, we are currently pretraining the AoT model on ImageNet-21K to further evaluate its performance. Once these results are obtained, we will include them in the revised manuscript. We hope this result can address your concern.

---

> > > > ### Author Response · Authors · 2024-12-02
> > > >
> > > > Dear Reviewer T5de,
> > > >
> > > > Thank you again for your thoughtful review and the time you have spent evaluating our work.
> > > >
> > > > As the author-reviewer discussion period will end soon, we are eager to know whether our responses have addressed your concerns. Your feedback is highly valuable to us, and if you have any remaining questions or require further clarification, we are more than happy to provide additional information. If our responses have resolved your concerns, we would sincerely appreciate it if you could consider updating your score.
> > > >
> > > > Authors.

---

### Official Review · Reviewer_Q2fD · 2024-11-03

**Soundness:** 1
**Presentation:** 3
**Contribution:** 2
**Rating:** 3
**Confidence:** 5

**Summary:**

This paper introduces a class of interpretable attention-only transformers that compresses noisy initial token representations into a mixture of low-dimensional subspaces, primarily building on the multi-head subspace self-attention introduced in [1]. The authors provide theoretical guarantees on the model's denoising performance, demonstrating that the signal-to-noise ratio improves with each layer. Experimental results in both language and vision tasks illustrate the advantages of the proposed transformer.

**Strengths:**

1. The paper provides a good review of the multi-head subspace self-attention from [1].

2. The paper is well-written with illustrative figures.

**Weaknesses:**

1. The proposed attention-only transformer is just a simple extension by unrolling Eqn. 3 in the paper, which is the multi-head subspace self-attention in [1]. Most heavy-lifting work has been done in [1], and unrolling Eqn. 3 is just an incremental contribution.

2. The motivation behind the proposed unrolling transformers is to denoise the noisy initial token representations towards a mixture of low-dimensional subspaces. However, it is not clear why denoising token representations should help improve the performance of the model. Also, there is no empirical evidence showing that the tokens are drawn from a mixture of noisy low-rank Gaussian distributions as assumed in Definition 1.

3. The experimental results are unconvincing.

* First, in Table 1, test perplexity should be used to evaluate the models rather than just validation loss and zero-shot accuracy. Also, in Table 1, the proposed models need many more parameters compared to the baseline GPT-2 while still worse than the baseline GPT-2 in most cases.

* Second, in Table 2, the performance of the proposed model, AoT, on ImageNet-1K is much worse than the baseline CRATE-\alpha model. For example, AoT-Huge has more parameters (86M) than the baseline CRATE-\alpha-B/16 but is 2% worse than the baseline in top-1 accuracy.

* Third, even though the authors claim that their proposed transformer is interpretable in the abstract and introduction, there are no experiments to show the interpretability. Similarly, one of the contributions mentioned in the introduction is to understand the role of self-attention and MLP layers, this is not really empirically discussed in the paper.

4. I highly doubt the efficiency of the proposed method. From Table 1 and Table 2, it is clear that the proposed model requires more parameters than the baseline transformer models. Also, the multi-head subspace self-attention in Eqn. 3 requires more computations compared to the baseline self-attention.

**Questions:**

1. Why shared-QKV subspace self-attention mechanisms, as mentioned in line 299?

2. Please provide the efficiency analysis, i.e., real-time running and memory usage, of the proposed transformer vs. the baseline transformer.

**References**

[1] Yaodong Yu, Sam Buchanan, Druv Pai, Tianzhe Chu, Ziyang Wu, Shengbang Tong, Hao Bai, Yuex- iang Zhai, Benjamin D Haeffele, and Yi Ma. White-box transformers via sparse rate reduction: Compression is all there is? arXiv preprint arXiv:2311.13110, 2023a.

**Details Of Ethics Concerns:**

I have no ethics concerns about the paper.

---

> ### Author Response · Authors · 2024-11-23
>
> We thank the reviewer for the careful review. In the following, we address the reviewer’s comments individually.
>
> **Q1.** The proposed attention-only transformer is just a simple extension by unrolling Eqn. 3 in the paper, which is the multi-head subspace self-attention in [1]. Most heavy-lifting work has been done in [1], and unrolling Eqn. 3 is just an incremental contribution.
>
> **A1.** We respectfully disagree with the comment that our contribution is incremental. While [1] introduces the MSSA mechanism, it mainly focuses on the derivation and conceptual introduction of the operation, without conducting a fine-grained theoretical analysis of its properties and behavior in deep architectures. Our work makes several significant advancements beyond [1]:
>
> **Rigorous Theoretical Analysis**: In our work, we provide a theoretical framework that rigorously proves the efficiency of the MSSA operator when unrolled into a deep transformer architecture. Specifically, we demonstrate that each layer achieves a linear improvement in the signal-to-noise ratio of token representations.
>
> **Architectural Innovation**: By unrolling Eqn. 3, we derive a highly compact attention-only transformer architecture that entirely omits the MLP layers, reducing redundancy while maintaining strong performance. This minimalistic design differs significantly from the architectures in [1], which do not explore such an approach.
>
> **Q2.** The motivation behind the proposed unrolling transformers is to denoise the noisy initial token representations towards a mixture of low-dimensional subspaces. However, it is not clear why denoising token representations should help improve the performance of the model. Also, there is no empirical evidence showing that the tokens are drawn from a mixture of noisy low-rank Gaussian distributions as assumed in Definition 1.
>
> **A2.** We would like to clarify the motivation and assumptions behind our approach:
>
> **Why denoising token representations should improve the model performance:**  The goal of representation learning is to find the underlying structure of data while effectively removing noise. By progressively denoising token representations toward a structured and compact form, the transformer architecture extracts cleaner and more informative features, leading to improved performance on downstream tasks. A mixture of low-dimensional subspaces serves as one specific example of such a structured and compact representation. However, it is worth noting that transformers can also adaptively transform token representations into other meaningful structures, depending on the nature of the data and the task at hand.
>
> **Empirical evidence showing that the tokens are drawn from a mixture of noisy low-rank Gaussian distributions:** Recent empirical studies on language tasks have raised the “linear representation hypothesis”, which posits that token representations can be linearly encoded as one-dimensional feature vectors in the activation space of LLMs [1], and the “superposition hypothesis”, which further hypothesizes that token representations are a sparse linear combination of these feature vectors [2]. These
> empirical studies demonstrate that the token representations lie on a union of (possibly many) low-dimensional subspaces.
>
> [1] Park, K., Choe, Y. J., & Veitch, V. (2023). The linear representation hypothesis and the geometry of large language models. arXiv preprint arXiv:2311.03658.
>
> [2] Elhage, N., Hume, T., Olsson, C., Schiefer, N., Henighan, T., Kravec, S., ... & Olah, C. (2022). Toy models of superposition. arXiv preprint arXiv:2209.10652.

---

> ### Author Response · Authors · 2024-11-23
>
> **Q3.** Comparison of the model performance between the proposed transformer architecture and the state-of-the-art ones
>
> **A3.** The main goal of our empirical studies is not to surpass the performance of state-of-the-art transformers. Instead, our objective is to demonstrate that the proposed interpretable and attention-only transformer can achieve performance close to that of standard transformers while maintaining the simplicity and interpretability of its design. This focus allows us to highlight the potential of our architecture as a theoretically grounded alternative to existing approaches, particularly in scenarios where interpretability is a priority. Moreover, we will use a grid search approach to optimize our hyperparameters and some other approaches to improve the model performance in our revised manuscript.
>
> **Q4.**  Even though the authors claim that their proposed transformer is interpretable in the abstract and introduction, there are no experiments to show the interpretability. Similarly, one of the contributions mentioned in the introduction is to understand the role of self-attention and MLP layers, this is not really empirically discussed in the paper.
>
> **A4.** **Interpretability of our proposed transformer:** We should clarify that interpretability in our work refers to understanding how the transformer architecture transforms token representations across layers. This notion differs from the traditional definition of interpretability in large language models (LLMs), which often emphasizes extracting concise, human-readable explanations for specific token representations. Our approach to interpretability is rooted in the mathematical and structural understanding of the transformer’s behavior, providing a clearer view of how information is processed and denoised at each layer.
>
> **Discussion of the role of self-attention and MLP layers:** We have proved the role of self-attention in Theorem 1 and discussed the role of MLP layers, such as Lines 481-482.
>
> **Q5.** Why shared-QKV subspace self-attention mechanisms, as mentioned in line 299?
>
> **A5.** Because our studied MSSA in Eq (5) can be viewed a special instance of standard self-attention formula by setting $Q=K=V=U_k$.
>
> **Q6.** Please provide the efficiency analysis, i.e., real-time running and memory usage, of the proposed transformer vs. the baseline transformer.
>
> **A6.** Thanks for pointing this out. We have conducted the experiments and reported the Gflops, GPU memory, and running time per pass in Table 3 in Appendix B.0.3.

---

### Official Review · Reviewer_9ZDx · 2024-11-04

**Soundness:** 3
**Presentation:** 3
**Contribution:** 2
**Rating:** 5
**Confidence:** 3

**Summary:**

Paper proposes an attention only transformer with guarantees that each subsequent layer improves the SNR of the signal. they provide theoretical insights in to how such a system is beneficial and also run some baseline experiments on how this holds up against traditional transformer networks.

**Strengths:**

The approach is very good and sound. builds on strong pre-existing work and adds an important piece to it. The flow of the paper is good and adds theoretical justifications on attention only transformers.

**Weaknesses:**

The paper doesn't offer any practical analysis on the theory. The results on language and vision are good, but using the paper's own arguments, the training dynamic of such systems and modern frameworks can reach convergence outside of this papers contributions. The key piece missing is an approach to understanding how the proofs of the theory play out in actual training.

**Questions:**

How does the reformulation in the paper for AoT converge?
How can we observe better SNR in these models?

---

> ### Author Response · Authors · 2024-11-20
>
> **Q1.** The paper doesn't offer any practical analysis on the theory. The results on language and vision are good, but using the paper's own arguments, the training dynamic of such systems and modern frameworks can reach convergence outside of this papers contributions. The key piece missing is an approach to understanding how the proofs of the theory play out in actual training.
>
> **A1.**  Thank you for your insightful comment. To clarify, our main theoretical result does not pertain directly to the training process of transformers. Instead, it demonstrates that under the proposed network architecture, we show that each layer of the transformer denoises token representations at a linear rate.
>
> If we understand correctly, your concern is whether the proven linear denoising performance of the studied transformers can be observed when they are trained on real-world datasets (please let us know if our understanding is wrong). This theoretical result holds when the token representations approximately satisfy a mixture of low-rank Gaussians. Notably,  any distribution can be well approximated by multiple Gaussian distributions [1]. In this sense, our result provides a meaningful foundation for understanding the behavior of the architecture in practical settings.
>
> [1] Borkar, V. S., Dwivedi, R., & Sahasrabudhe, N. (2016). Gaussian approximations in high dimensional estimation. Systems & Control Letters, 92, 42-45.
>
> **Q2.** How does the reformulation in the paper for AoT converge? How can we observe better SNR in these models?
>
> **A2.** We apologize that we are not entirely clear on the first question. It would be great if you could provide more details about the **reformulation for AoT converge**.
>
> According to Eq (8) in Theorem 1, when the step size $\eta$ or the thresholding parameter $\tau$ is large, the SNR increases at a faster rate as the number of layers grows. Additionally, as the number of layers increases, the SNR at the final layer is further improved.

---

> > ### Comment · Reviewer_9ZDx · 2024-11-26
> > **Clarification for question 2**
> >
> > The question here is related to question 1. You show through mathematical proof that SNR should improve for attention only transformer (AoT). Can you also show this in converged models? Is there a notion of signal and noise that can be measured and and shown to improve layer on layer

---

> > > ### Author Response · Authors · 2024-11-26
> > >
> > > Thanks for your clarification. Thank you for the clarification. Yes, we can empirically demonstrate this improvement in practical settings. The procedure is as follows: (1) Apply **subspace clustering algorithms** to the token representations at each layer to identify the basis matrices and the associated noise components; (2) Compute the signal-to-noise ratio (SNR) for each layer based on the identified subspaces and noise components; (3) Visualize the SNR show how it improves layer by layer. We will add these experiments in our revised manuscript.

---

### Official Review · Reviewer_aKa8 · 2024-11-04

**Soundness:** 3
**Presentation:** 3
**Contribution:** 3
**Rating:** 6
**Confidence:** 4

**Summary:**

This paper presents a novel approach to transformer architecture by proposing an attention-only model that utilizes multi-subspace self-attention (MSSA) for efficient denoising of token representations. The authors demonstrate that their unrolled optimization framework simplifies transformer architectures significantly, achieving performance comparable to traditional models while enhancing interpretability. Rigorous theoretical analysis supports the denoising capabilities of their architecture, showing improvements in signal-to-noise ratio across layers. Extensive empirical evaluations across language and vision tasks validate the practical effectiveness of their proposed design.

**Strengths:**

The use of unrolled optimization to design a simplified transformer architecture presents a fresh perspective on transformer design, focusing on interpretability and efficiency.

The paper rigorously proves the denoising effectiveness of the proposed architecture, providing a solid mathematical foundation for its claims.

The authors conduct comprehensive experiments across diverse tasks, supporting the practical viability of their approach and demonstrating competitive performance.

**Weaknesses:**

One notable weakness of this study is the lack of validation on state-of-the-art transformer architectures, such as Llama, and benchmark tasks, including MMLU. It would strengthen the paper significantly to include evaluations on these platforms to better assess the proposed method's performance in comparison to leading models in the field.

The paper could benefit from more detailed comparisons to the latest transformer variants such as mamba and linear transformers, which may also claim reduced complexity or improved performance.

**Questions:**

See the weaknesses above.

---

> ### Author Response · Authors · 2024-11-20
>
> **Q.** Validation on state-of-the-art transformer architectures, such as Llama, and comparisons to the latest transformer variants such as mamba and linear transformers
>
> **A.** Thank you for the suggestion. We would like to clarify that the main goal of our empirical studies is not to surpass the performance of state-of-the-art transformers. Instead, our objective is to develop a minimalistic transformer-like deep architecture consisting of fully interpretable and provably effective layers that can achieve performance close to that of standard transformers.
>
> That said, we are currently conducting experiments to validate the performance of our transformer on state-of-the-art models such as Llama and also compare our architecture with leading transformer variants such as Mamba across various benchmark tasks, as suggested. Due to limited computing resources, we aim to update these results before the rebuttal deadline. Furthermore, we will include these results in our paper to demonstrate the performance of our proposed architecture.
>
> Updated: We have compared the in-context learning performance of our proposed architecture with Llama architectures in Figure 7 in the revised manuscript. It is observed that the proposed architecture achieves comparable performance to Llama.

---

### Author Response · Authors · 2024-11-23
**Global response**

We thank all the reviewers for their insightful comments and constructive feedback. To summarize, our work introduces an attention-only transformer architecture that is both interpretable and effective. We rigorously demonstrate that each layer of our proposed transformer denoises token representations at a linear rate, given initial token representations sampled from a mixture of low-rank Gaussians. Our contributions aim to bridge theoretical insights with empirical validation. Below, we address two main points raised in the reviews:

- **Comparison to the state-of-the-art transformers:** The main goal of our empirical studies is not to surpass the performance of state-of-the-art transformers. Instead, our objective is to demonstrate that the proposed interpretable and attention-only transformer can achieve performance close to that of standard transformers while maintaining the simplicity and interpretability of its design. This focus allows us to highlight the potential of our architecture as a theoretically grounded alternative to existing approaches, particularly in scenarios where interpretability is a priority.

- **Difference of the interpretability between our work and the existing works:**  In our work, interpretability refers to understanding how the transformer architecture transforms token representations across layers. This notion differs from the traditional definition of interpretability in large language models (LLMs), which often emphasizes extracting concise, human-readable explanations for specific token representations. Our approach to interpretability is rooted in the mathematical and structural understanding of the transformer’s behavior, providing a clearer view of how information is processed and denoised at each layer.

---

### Comment · Area_Chair_saGL · 2024-11-25
**Author Reviewer Discussion**

Dear Reviewers,

Thank you for your efforts in reviewing this paper. We highly encourage you to participate in interactive discussions with the authors before November 26, fostering a more dynamic exchange of ideas rather than a one-sided rebuttal.

Please feel free to share your thoughts and engage with the authors at your earliest convenience.

Thank you for your service for ICLR 2025.

Best regards,

AC

---

> ### Author Response · Authors · 2024-11-26
>
> Dear Reviewers,
>
> Thank you again for your thoughtful review and the time you have spent evaluating our work.
>
> As the author-reviewer discussion period approaches its conclusion, we noticed that we have not yet received a follow-up response from you. We are eager to know whether our responses have addressed your concerns. Your feedback is highly valuable to us, and if you have any remaining questions or require further clarification, we are more than happy to provide additional information.
>
> If our responses have resolved your concerns, we would sincerely appreciate it if you could consider updating your score.
>
> Thank you for your time and consideration.
>
> Authors.

---

### Meta-Review · Area_Chair_saGL · 2024-12-20

**Metareview:**

This submission proposes "attention-only" transformers, which have so-called multi-subspace self-attention (MSSA) layers. The resulting transformer architecture is very compact and does not require the usual MLP layers. The paper has receives critical and borderline reviews. The reviewers acknowledge the benefit of the simplified architecture and the solid mathematical background. One important criticism is the lack of comparison to state of the art transformer architectures (e.g. aKa8, Q2fD), the novelty of the work (e.g. Q2fD), interpretability (T5de) and the experimental validation in terms of practical analysis (9ZDx). An additional evaluation on llama was provided during the rebuttal but the concerns regarding novelty and insufficient experimental validation still persist after the rebuttal, leaving the paper with a borderline reject score.

**Additional Comments On Reviewer Discussion:**

Reviewers Q2fD, T5de and 9ZDx have actively engaged in discussion with the authors. However, their concerns, in particular with respect to the experimental validation could not be resolved during the rebuttal phase.

---

### Decision · Program_Chairs · 2025-01-22

Reject